# The Dysferlinopathies Conundrum: Clinical Spectra, Disease Mechanism and Genetic Approaches for Treatments

**DOI:** 10.3390/biom14030256

**Published:** 2024-02-21

**Authors:** Saeed Anwar, Toshifumi Yokota

**Affiliations:** Department of Medical Genetics, Faculty of Medicine and Dentistry, University of Alberta, Edmonton, AB T6G 2R3, Canada; sanwar@ualberta.ca

**Keywords:** dysferlinopathy, limb-girdle muscular dystrophy recessive type 2 (LGMDR2), Miyoshi myopathy, distal myopathy with anterior tibial onset (DMAT), dysferlin, membrane resealing, genetic therapy, mini-dysferlin, exon skipping

## Abstract

Dysferlinopathies refer to a spectrum of muscular dystrophies that cause progressive muscle weakness and degeneration. They are caused by mutations in the *DYSF* gene, which encodes the dysferlin protein that is crucial for repairing muscle membranes. This review delves into the clinical spectra of dysferlinopathies, their molecular mechanisms, and the spectrum of emerging therapeutic strategies. We examine the phenotypic heterogeneity of dysferlinopathies, highlighting the incomplete understanding of genotype-phenotype correlations and discussing the implications of various *DYSF* mutations. In addition, we explore the potential of symptomatic, pharmacological, molecular, and genetic therapies in mitigating the disease’s progression. We also consider the roles of diet and metabolism in managing dysferlinopathies, as well as the impact of clinical trials on treatment paradigms. Furthermore, we examine the utility of animal models in elucidating disease mechanisms. By culminating the complexities inherent in dysferlinopathies, this write up emphasizes the need for multidisciplinary approaches, precision medicine, and extensive collaboration in research and clinical trial design to advance our understanding and treatment of these challenging disorders.

## 1. Introduction

Dysferlinopathies represent a phenotypically heterogeneous spectrum of muscular dystrophies with autosomal recessive inheritance, characterized by abnormal amyloid deposition and fragments of dysferlin within skeletal muscle tissues [1,2]. With a prevalence estimated between 1 in 14,000 and 1 in 2 million, dysferlinopathies are one of the most common forms of adult-onset muscular dystrophies [3,4,5]. This spectrum of conditions is secondary to mutations in the *DYSF* (dystrophy-associated fer-1-like) gene located on chromosome 2p13, which encodes the dysferlin—a protein crucial for skeletal muscle membrane repair [2,6,7,8,9]. Mutations in the *DYSF* gene precipitate either a complete loss or a functional impairment of dysferlin [1,2,6].

The protein dysferlin is integral to muscle physiology. Located within the complex molecular structure of the transverse-tubule (T-tubule) network, it maintains the sarcolemmal matrix’s stability and integrity [8,9,10,11,12]. Structurally, it is composed of seven lipid-responsive C2 domains, an inner DysF (iDysF) domain, two Fer domains, and a transmembrane anchor—featuring a very sophisticated architecture [9,13,14,15,16,17,18,19,20]. Interactome studies reveal that dysferlin has synergistic interactions with proteins, e.g., affixin, caveolin-3, and calpain-3, which are all crucial for muscle membrane repair and integrity, thus broadening its role in muscle cell function [21,22,23]. Dysferlin is almost ubiquitously expressed but is most abundant in muscular structures, e.g., skeletal and cardiac muscles [9,24]. Dysferlin’s transcriptomic markers can be detected in diverse anatomical regions, e.g., bone marrow, monocytes, liver, brain, thyroid, lung, endothelium, testes, pancreas, kidneys, and human placenta [25,26,27].

Clinically, dysferlinopathies manifest in multiple forms, including Miyoshi myopathy (MM), limb-girdle muscular dystrophy recessive type 2 (LGMDR2, also known as limb-girdle muscular dystrophy type 2B or LGMD2B), and the less common distal myopathy with anterior tibial onset (DMAT) [2,6,22,28,29,30,31]. Onset is typically in adolescence or early adulthood, with significant difficulties including tiptoe standing or stair climbing [4,23,31,32,33,34]. While muscle weakness tends to progress slowly over years, some rare cases report a rapid decline in mobility within approximately 5 years [34]. Modern medical modalities—including MRI, electromyography, and histopathological investigations—provide important insights into muscle fiber variations, necrosis, regeneration, and the accumulation of fat and connective tissue [23].

Historically, although dysferlinopathies have been under study since their initial documentation in Japan in the 1970s, our understanding of the molecular intricacies of these conditions remains elusive, largely due to the incomplete understanding of dysferlin’s core functions [22,35,36,37,38]. Nonetheless, contemporary breakthroughs, especially the development of dysferlin-deficient mouse models, offer important insights and spearhead potential therapeutic strategies [22,23,35]. Emerging therapeutic approaches span from anti-sense-mediated exon skipping to myoblast transplantation and avant-garde gene therapies.

This article aims to provide an overview of the current understanding of dysferlinopathies, from their biology and clinical implications into therapeutic avenues. It explores the current clinical management paradigms, discusses the latest pharmaceutical advances targeting this disease spectrum, and spotlights knowledge gaps for future research endeavors.

## 2. Dysferlinopathies: Clinical Landscape and Phenotypic Variability

Dysferlinopathies present a multifaceted clinical landscape, displaying varied patterns of muscular involvement (Table 1). These conditions can be differentiated by their distinct patterns of muscle weakness at the onset [29,31,34,39]. LGMDR2 is primarily characterized by proximal muscle weakness, with the earliest signs of weakening and atrophy appearing in the pelvic and shoulder girdle muscles. MM predominantly impacts the distal muscles—originating in the gastrocnemius and soleus muscles of the calf and plantar regions, it progresses toward the thigh and gluteal muscles. DMAT, on the other hand, specifically affects the upper portion of the tibial muscles in the lower limbs. Interestingly, even within the same family, individuals can exhibit different patterns of muscle weakness [40,41]. This variability, combined with the overlapping proximal and distal muscle involvements, can sometimes blur the distinction between LGMDR2 and MM. Consequently, clinicians have identified an intermediate clinical category known as the dysferlin-deficient proximo-distal phenotype [42].

A hallmark feature of dysferlinopathies is a sharp increase in serum creatine kinase (CK) levels, indicative of muscle damage [43,44,45]. During the initial stages of the disease, CK levels skyrocket to reach 50–200 times typical values [43]. The heightened CK levels, if not progressively escalating, persist throughout the course of the disease. Exceptionally, in cases identified as dysferlin-associated asymptomatic hyperCKemia, a significant rise in serum CK is the only detectable symptom, leaving the muscles largely unaffected [30]. Elevated CK levels along with muscle weakness, particularly in distal regions, historically served as a key diagnostic criterion for diagnosing most of the recessive muscular dystrophies, and dysferlinopathies were no exception [22,31,43]. However, with advancements in diagnostic modalities, the reliance on CK levels as a diagnostic metric for dysferlinopathy has receded [22,23,43].

Once affected, certain muscles, e.g., the gastrocnemius, soleus, and subscapularis, are especially vulnerable in dysferlinopathies, while others like the elevator scapulae are less affected; the reason underneath is not well understood though [23,55]. While muscle involvement is generally symmetrical, some patients might exhibit asymmetrical muscle degradation. MRI studies highlight an increase in muscles’ fat content, especially in non-ambulant patients, with the quadriceps frequently being severely affected [23,58]. In contrast, the contractile area usually diminishes [58].

Misdiagnosis is a concern with dysferlinopathies due to symptomological overlaps with conditions like polymyositis (PM), other limb-girdle muscular dystrophies (LGMDs), and even Charcot–Marie–Tooth disease (CMT) [43,50,58,59]. The distinction lies in the absence of elevated CK levels and specific sarcolemma upregulation in CMT [43,59]. Electromyographic (EMG) studies, complemented by CT and MRI, aid in differentiating dysferlinopathy from CMT [59]. While a Western blot determines the presence or absence of dysferlin protein in tissues, genetic screening, complemented by advanced sequencing, is necessary to pinpoint specific mutations [31,59,60,61,62]. In settings with limited resources, polymerase chain reaction (PCR) and array comparative genomic hybridization (CGH) can offer crucial DNA-level diagnostic insights [63].

### 2.1. LGMDR2

Clinically, LGMDR2 typically begins to manifest symptoms between ages 13 and 40, illustrating a slow yet inexorable progression in muscular function—a trend corroborated in multiple studies encompassing diverse ethnic and demographic cohorts, especially in regions where consanguinity is prevalent [1,28,40,64,65,66,67,68,69,70].

The disease initially weakens the proximal regions of the thigh muscles, leaving shoulder muscles relatively unscathed for prolonged periods. As it advances, the distal leg muscles deteriorate, and common complaints include fatigue, stair-climbing difficulties, and general muscle weakness [28,33,34]. An archetypal gait pattern, termed the “dysferlin gait”, stands out as a hallmark for LGMDR2 patients. This unique walking pattern stems from the debilitated quadriceps (thigh muscle), altering knee movement during walking. Compensation leads to a wider stride, though its length remains unchanged. Notably, some patients face simultaneous weakening in both leg muscle groups from the onset. At the advanced stage, 15–50% of patients may lose ambulation completely and become wheelchair-bound [65,67].

Histological evaluations of muscle biopsies reveal varied muscle fiber sizes, increased connective tissues, fatty deposits, and occasional inflammation, necrosis, and fibrotic changes [7,40,64,65,67,70]. Early stages might exhibit minimal structural changes but showcase regeneration signs, e.g., fibers with central nuclei. Ultrastructural studies highlight membrane damage, evident through subsarcolemmal vesicles and vacuoles. In some cases, CT scans detect subtle damage in distal muscles [22,43].

### 2.2. MM

MM, classified as a distal myopathy, primarily affects those aged 14 to 40, with initial symptoms manifesting in the calf muscles [7,33,44,45,68,69]. This leads to challenges in walking and stair climbing. Unique to MM, calf discomfort and pain differentiate it from other distal muscular dystrophies [22,68].

Histological analysis of muscle biopsies from MM patients reveals varying histological changes based on the muscle group [44,45,68,71,72,73,74], which often mirror the histologic features of dystrophinopathies. Atrophic calf muscle biopsies typically show significant fibrosis and fat replacement, leading to a marked loss of muscle fibers [73,74]. In contrast, quadriceps muscle biopsies often present minimal myopathic alterations. The hamstring muscle (specifically, the biceps femoris) may display intermediate changes, evident by varied fiber sizes and the presence of necrotic and regenerating fibers [72]. While vacuoles are uncommon in MM, sporadic cases report their marginal presence, making them a non-specific histological sign. A strikingly elevated blood serum CK level serves as a hallmark for MM diagnosis [31,75]. For MM patients, CK levels usually soar to 50–200 times above the standard range.

### 2.3. DMAT

DMAT is the rarest of the three major phenotypes within the dysferlinopathies spectrum [2,22,29,31]. It manifests between ages 14 and 30 and is marked by leg weakness, specifically in the anterior compartment of the leg, leading to foot drop [2,29,31,76]. It progresses faster than LGMDR2 and MM, affecting both lower and upper proximal muscles. Patients typically become wheelchair-bound within 10–22 years after onset [2,22,29]. While cranial muscles remain unaffected, serum CK levels soar 20 to 70 times above normal [29]. Muscle biopsies indicate moderate myopathic changes but lack vacuoles [29,76]. While DMAT’s onset in the anterior tibial muscles sometimes mirrors Nonaka myopathy, it is differentiated by its elevated CK levels and absence of vacuoles in muscle studies.

### 2.4. Dysferlin-Deficient Proximo-Distal Phenotype

Dysferlin-deficient proximo-distal phenotype occupies a unique position in the spectrum of dysferlinopathies, nestled between LGMDR2 and MM [7,22,31,42,77]. Exhibiting traits from both conditions, its dual nature complicates phenotypic classification [31,42,77]. This phenotype presents a simultaneous onset of weakness in both the distal and proximal muscles, blurring the lines between LGMDR2 and MM.

Symptoms appear in the early twenties to thirties and vary widely [22,31,42,43]. Although the disease progression tends to be gradual, it can follow symmetrical or asymmetrical trajectories, occasionally focusing on specific muscle regions. A first-line indicator is elevated serum CK levels, rising 10–50 times above normal.

Recent studies involving MRI technology have provided interesting insights. Adding further to the intricacies of phenotypic classification within the spectrum, these studies have revealed that all patients with dysferlinopathies with muscular involvement exhibit both proximal and distal muscle involvement on imaging, irrespective of their outward symptoms pointing to LGMDR2 or MM [77,78]. Thus, the subtle distinctions among LGMDR2, MM, and the proximo-distal phenotype remain an area of intense scholarly debate.

### 2.5. Asymptomatic HyperCKemia

Dysferlin-associated asymptomatic hyperCKemia, often termed isolated hyperCKemia, stands out as a unique and rare clinical variant within the spectrum of dysferlinopathies [30,31]. This condition presents a pronounced elevation in serum CK levels but little to no overt muscle involvement. This elevated CK level, though seemingly isolated, may be indicative of underlying muscle pathology at the cellular level, hinting that clinical symptoms could surface over time [31].

As this condition typically acts as a presymptomatic harbinger, many individuals with elevated CK levels will, in due course, manifest muscle weakness and atrophy [22,30,31]. Occasionally, these individuals might experience calf muscle enlargement or hardening, adding a layer of diagnostic complexity [31]. Such presentations could easily be mistaken for dystrophinopathies, specifically Becker muscular dystrophy [31]. Given its subtle and deceptive onset, recognizing dysferlin-associated asymptomatic hyperCKemia is pivotal for appropriate and routine clinical intervention.

### 2.6. Dysferlin-Associated Congenital Phenotype

Dysferlin-associated congenital phenotype is extremely rare. Affected individuals often exhibit delayed head control and pervasive hypotonia. Additionally, they struggle with limb weakness and difficulties in walking, running, and climbing stairs—symptoms that are emblematic of dysferlinopathies [79]. Moreover, they have elevated serum CK levels.

## 3. Dysferlin: A Mosaic of Uncharted Roles and Functionalities

In the late 1990s, an international research group, led by Dr. Robert H. Brown Jr. from Massachusetts General Hospital, uncovered the *DYSF* gene associated with dysferlinopathies [2]. Around the same time, Dr. Catherine Bushby’s team based in the UK also made significant contributions to the foundational understanding of the disease [1]. The *DYSF* gene, which codes for the dysferlin protein, is located on chromosome 2p13 and occupies approximately 220 kb in the genome, making it one of the most extensive genes in the human genome [6,80]. The canonical transcript of *DYSF* consists of 55 exons, resulting in a coding sequence of around 6.2 kb and translating into a 2081 amino acid long protein (Figure 1) [1,2,6,80]. Furthermore, at least 15 alternative transcripts of *DYSF* are known [80]. While *DYSF* was initially identified to express in the sarcolemma and cytoplasmic vesicles of skeletal and cardiac muscle cells, subsequent studies have revealed its presence in various other cells and tissues, including the brain, lung, immune cells, and even placenta [1,2,27,61,81].

### 3.1. Into the Structural Intricacies of Dysferlin

Dysferlin, a colossal 237-kilodalton protein, is a member of the fer-1 (fertility factor 1-like proteins in *Caenorhabditis elegans*) family and is involved in vesicle fusion and membrane trafficking [1,2,82,83]. Alongside dysferlin, myoferlin is another notable member of the ferlin family, sharing structural similarities and functional roles [84,85,86]. Myoferlin is primarily known for its involvement in muscle cell membrane fusion and repair, akin to dysferlin, but it also plays a distinct role in myoblast fusion and muscle regeneration [87,88]. The interplay and differences between dysferlin and myoferlin in muscle physiology underscore the complexity and diversity within the ferlin family.

The ferlin family of proteins, encompassing both dysferlin and myoferlin, is characterized by four to seven C2 calcium/phospholipid-binding domains, essential for intracellular signalling and regulation, and a C-terminal transmembrane domain [86,88]. Additionally, these proteins contain a distinctive DysF domain, contributing to their multifaceted role in cellular physiology [89]. Ferlins are crucial in processes like vesicle trafficking and membrane fusion, especially in muscle and nerve cells, playing key roles in membrane repair, muscle development, and synaptic vesicle trafficking.

Dysferlin features an extensive cytoplasmic N-terminal domain, a C-terminal transmembrane domain, and a brief extracellular domain, giving it a cytosol-facing architecture (Figure 2) [89,90,91]. The cytosolic segment contains seven conserved C2 domains, each perceived to be performing a specific operation in dysferlin’s functionality [13,91,92,93,94,95,96]. Furthermore, the protein harbors Fer (ferlin) and DysF domains, which, when mutated, can be the harbingers of muscular dystrophies [89,90,91,97,98].

The seven C2 domains, labeled C2A through C2G, are particularly significant in dysferlin’s architecture. Each of these domains, except C2B, is involved in voltage-induced calcium regulation, and they display unique evolutionary lineages compared to other members of the ferlin family, suggesting specialized functions [13,91,92,93,94,95,101]. Notably, the C2A domain undergoes conformational changes upon calcium binding, highlighting its significant role in calcium-mediated activities [96,99,101,102]. Recent studies indicate that the deletion of either the C2A or C2B domain completely hinders membrane repair in cell models [101]. Moreover, deletion of C2C–C2G domains induces a partial repair deficiency, which intensifies without extracellular Ca^2+^. This also significantly alters Ca^2+^ dynamics and predominantly assigns the protein to the endoplasmic reticulum. Lacking only the C2A domain allows dysferlin’s proper localization to T-tubules; however, it concurrently impairs Ca^2+^ signaling during osmotic challenges. Bar C2B, every C2 domain influences Ca^2+^ signaling, with C2C to C2G particularly determining the Ca^2+^ dependency of the repair mechanism. Collectively, dysferlin’s C2 domains play pivotal roles in T-tubule targeting, membrane repair facilitation, and calcium regulation.

Apart from the C2 domains, dysferlin showcases the FerA and DysF domains. Intriguingly, the DysF domain displays an embedded structure: one DysF domain (known as inner DysF or iDysF domain) is located within another [89,97]. This nested structure is a product of gene duplication events [97,98]. While the FerA domain’s role in calcium-dependent interactions with membranes is documented, the exact functionality of the iDysF domain remains a subject of ongoing research [86,89,97,103]. Dysferlin’s architecture also includes an extensive intracellular cytoplasmic N-terminal domain [104]. At the extreme end, it houses a C-terminal transmembrane domain, which anchors the protein to the membrane, and a brief extracellular C-terminal domain. This orientation places a significant portion of dysferlin in the intracellular milieu, offering opportunities for interactions with myriad intracellular proteins and signaling molecules.

The incredible multidomain architecture of dysferlin has direct functional implications. The C2 domains, especially, enable the protein to associate swiftly with lipid membranes under conditions of elevated intracellular calcium [13,96,101]. This rapid association is a testament to dysferlin’s role in membrane repair, particularly after muscle injuries [14,93,99]. Furthermore, while a small part of dysferlin interacts with the external environment, most of the protein is inside the cell due to the positioning of its transmembrane domain. This internal part, especially the C2 domains, could be pivotal for the intracellular processes it mediates, ranging from vesicle trafficking to calcium signaling.

Dysferlin’s structure is not just an intricate assembly of domains; it is a masterclass in functional design [89,97]. Each domain, fold, and molecular interaction is vital for its role in muscle physiology. While our understanding has grown exponentially over the past decades, there remain mysteries waiting to be unraveled about this fascinating protein. As researchers continue to delve deeper into dysferlin’s structure–function relationships, they inch closer to unlocking therapeutic potentialities for dysferlin-associated disorders [23,35,105,106].

### 3.2. Multifaceted Functional Aspects of Dysferlin

Dysferlin, anchored to the sarcolemmal membrane, T-tubule vesicle, and intracellular vesicles in the muscle cells, mediates a plethora of functions, encompassing membrane repair, calcium regulation, and inflammatory modulation (Figure 3) [17,82]. It aids in shuttling intracellular vesicles to injury sites, effectively “patching” and resealing the membrane [8,9,82]. Within the T-tubule network, dysferlin’s association with calcium-regulating proteins is crucial for muscle contraction [10,11,12,57].

While dysferlin presents a distinct immune signature, the nuances of its interaction with immune and inflammatory components remain uncharted [15,107,108]. Dysferlin deficiency also correlates with specific lipid-related pathologies, impacting metabolism and lipid composition [13]. A comprehensive understanding of the intricate nature of dysferlin within the broader biological network demands further in-depth research.

#### 3.2.1. Dysferlin in Membrane Repair and Integration

Skeletal muscles, known for adaptability, recalibrate their structure and function in response to physiological needs. Mechanical stress from muscle activity can instigate microlesions in the plasma membrane, causing a surge of calcium influx, prompting vesicle fusion at the injury site, central to which is dysferlin [8,9,82,106]. Dysferlin enables the precision transport of vesicles to injury zones and is instrumental in vesicle fusion, collaborating with membrane stability proteins, e.g., annexins and SNARE (soluble N-ethylmaleimide-sensitive factor attachment protein receptors) [9,109]. Importantly, dysferlin may also act as a vital Ca^2+^ donor during fusion. The protein mitsugumin 53 (MG53), crucial for vesicle transport in skeletal muscle cells, exhibits a unique interaction with dysferlin, underscoring the importance of its synergistic roles [17,100]. Furthermore, dysferlin’s association with caveolin-3 (Cav3) and myoferlin suggests a collaborative protein complex vital for muscle health [84,85]. In addition, it has been reported that dysferlin-deficient myoblasts show reduced levels of myogenin, resulting in a compromised fusion index [110].

Evaluation of dysferlinopathic muscles reveals marked abnormalities, particularly in dysferlin expression and membrane integrity. Both human and murine models underscore dysferlin’s quintessential role in membrane repair, despite the mechanisms being not fully elucidated.

#### 3.2.2. Calcium Regulation

Dysferlin plays an instrumental role in calcium handling, which is pivotal for muscle contraction and enzyme regulation that, in turn, can cause muscle damage [57]. The T-tubular system, quintessential to these calcium dynamics, is significantly influenced by dysferlin, leading to muscle weakness and fatigability [111,112,113,114].

Dysferlin’s spatial localization is within the T-tubule membranes, where it associates with a suite of proteins, including the dihydropyridine receptor (DHPR), caveolin-3, MG53, annexin A1, ryanodine receptor 1 (RyR1), and AHNAK [10,12,19,112,115,116]. Dysferlin-deficient muscles display aberrant T-tubule morphology, underscoring the importance of the protein in T-tubule conservation and highlighting increased susceptibility to external stressors [10,115,117]. Kerr and colleagues have postulated dysferlin’s integral role in maintaining calcium balance, especially under mechanical stress [10]. It is suggested that diltiazem, a dihydropyridine receptor (DHPR) antagonist, has a protective effect on dysferlin-deprived fibers when under osmotic stress. Complementing this, Demonbreun and colleagues proposed dysferlin’s putative involvement in the formation and remodeling of the T-tubule system, suggesting an association with core cellular vesicular transport and membrane fusion processes [117,118].

#### 3.2.3. Dysferlin in Immune and Inflammatory Response

Dysferlinopathies manifest a marked inflammatory signature, indicative of disrupted immune responses in dysferlin-deficient muscles [107]. These disruptions involve altered macrophage function, complement activation, and heightened oxidative stress [60,108,119,120,121,122,123,124,125,126]. Additionally, dysferlin’s presence in monocytes and its deficiency, which increases aggression and phagocytosis, might accelerate disease progression [119,127]. This highlights the delicate balance of immune cell infiltration in dysferlinopathy patients. Furthermore, dysferlin’s involvement in the secretion of cytokines and chemokines suggests a link between it and the Rho family of small GTPases, which regulate receptor-mediated phagocytosis and cytokine secretion [22,128]. Dysferlin deficiency might activate inflammation and fibrosis and lead to significant muscle fiber damage and dysfunction, emphasizing its crucial role in immune response dynamics [107,108,119]. In addition, dysferlin-deficient myoblasts showcase hindered secretion of cytokines like MCP-1 (monocyte chemoattractant protein 1) and reduced recycling rates of receptors such as transferrin and insulin-like growth factor (IGF) [117,129].

The exact Implications of dysferlin’s interactions with inflammatory/immune cells and proteins are not yet clear; however, there is a strong indication of an immune contribution to the dysferlinopathy pathology.

#### 3.2.4. Dysferlin in Cholinergic Signaling

A novel dimension of dysferlin’s functionality emerges in cholinergic signaling regulation [130]. Dysferlin-deficient mice display muscle strength loss and a cholinergic deficit, as shown by a decreasing muscle action potential frequency after sustained neural stimulation [130]. These findings, underscored by the restorative effects of the acetylcholinesterase inhibitor pyridostigmine bromide, hint at the significant pathophysiological role of cholinergic signaling regulation in pathology.

#### 3.2.5. Dysferlin beyond Skeletal Muscle

As mentioned earlier, dysferlin occurs in tissues beyond the skeletal muscle, e.g., the brain, heart, liver, lungs, endothelium, peripheral blood monocytes, testes, and placenta [25,26,27,131]. Dysferlin’s expression in monocytes aligns with its skeletal muscle levels across both healthy individuals and dysferlinopathy patients [61,132]. Dysferlin-deficient monocytes display heightened motility, suggesting its role in focal adhesion and cellular interaction [25]. Interestingly, high dysferlin levels have been identified in human placental trophoblasts, which transform into a large cell layer known as syncytiotrophoblasts (STB), essential for nutrient transport 81]. A compromised STB-apical membrane has implications in pre-eclampsia, a pregnancy disorder [133,134,135,136]. It is observed that dysferlin expression diminishes in pre-eclamptic placentas, emphasizing its potential role in placental membrane repair [135,136].

#### 3.2.6. The Dysferlin Interactome

A quarter-century has passed since dysferlin was first identified in the human genome; however, our understanding of its protein complex interactions remains in its infancy, with many of its binding partners yet to be fully understood. Dysferlin does not seem to be a crucial regulator of the dystrophin–glycoprotein complex (DGC) [8,35,137]. Dysferlin-deficient mice further support this, as they show no DGC alterations, suggesting dysferlin has distinct interactions and functions. Dysferlin activity can be observed in adult tissues and as early as 5–6 weeks in human embryonic development, primarily in the muscle fibers’ intracellular network and plasma membrane [24]. The above discussion sheds some light on the interactions dysferlin is involved in; however, dysferlin is suggested to have direct or indirect interaction with many other partners [21,138].

Current knowledge reveals that proteins interacting with dysferlin are implicated in a broad spectrum of physiological processes. These include membrane maintenance with proteins like annexin 1, annexin 2, affixin, and caveolin 3; cytoskeleton regulation through calpain 3 and AHNAK; and membrane repair, notably with MG53 [15,17,139,140,141,142,143,144,145]. The interaction of dysferlin with caveolin 3 at the T-tubule network and sarcolemma interface suggests a possible chaperoning role of caveolin-3 in the plasma membrane [139,144,145]. Annexins I and II, phospholipid binders involved in Ca^2+^ channel formation, and affixin, a focal adhesion protein, interact with dysferlin in human skeletal muscles [15,140,141]. Furthermore, the calcium-dependent protease calpain 3 interacts with dysferlin, emphasizing its importance in muscle remodeling [142,143].

A particularly significant interaction is observed between dysferlin and AHNAK, a protein involved in regulating cell growth, reorganizing the actin cytoskeleton, shuttling structural molecules across subcellular compartments, migration, and conferring cellular elasticity [19,21,138]. It is suggested that AHNAK and dysferlin co-localize at the sarcolemma and may interact directly through dysferlin’s C2A domain [16,19]. In muscle samples from dysferlinopathy patients, both AHNAK and dysferlin levels are reduced, with AHNAK showing a notable decrease in the sarcolemma but remaining stable in blood vessels, indicating a muscle-specific reduction.

Extensive interactome studies have identified 32 potential dysferlin partners, including interactions with moesin (MSN) and polymerase I and transcript release factor (PTRF, cavin 1) in mouse heart samples [138]. Moesin functions as a connector between the cell membrane and the actin structure, aiding in cellular signaling and movement [146,147,148]. PTRF, involved in the formation of caveolae, tiny indentations in the cell membrane, further expands dysferlin’s interaction network [22,148]. Immunohistochemistry studies have broadened our understanding of dysferlin’s interactome, confirming associations with 18 additional proteins [21].

The multifaceted role of dysferlin provides important insight into muscle health, calcium dynamics, immune response, and cholinergic signaling. Its deficiency triggers various disorders, emphasizing the need for deeper exploration of its workings and potential treatments [22,23,35].

### 3.3. Dysferlin in the Pathobiology of Dysferlinopathies

Research into the pathomechanism of dysferlinopathies predominantly focuses on dysferlin’s integral role in membrane repair. Deficits in membrane resealing have been observed in specific dysferlin-null tissues, as evidenced by both animal models and patient-derived cells, highlighting its importance beyond merely skeletal muscles [8,149,150].

While it might be presumed that correcting the membrane repair defect could reverse the dystrophic phenotype in dysferlinopathies, the reality proves to be more intricate. A notable instance involves a patient mutation that produced a quasi-dysferlin, which, restored membrane repair in dysferlin-deficient mice, but did not mitigate the distinctive dysferlinopathy-related muscle pathologies [151,152]. Similarly, overexpression of myoferlin in dysferlin-deficient specimens ameliorates sarcolemma resealing, yet the dystrophic manifestation remains [152]. This indicates that the pathomechanism of dysferlinopathy extends beyond a mere defect in plasma membrane repair.

Muscle health and function are heavily tied to intracellular membrane trafficking, central to the related processes, dysferlin’s interaction with other key proteins involved in vesicle trafficking (see Section 3.2) [16,153,154]. The clinical manifestations of dysferlinopathy are also marked by increased inflammation, potentially exacerbating the disease’s progression [108,124]. Despite the observed cellular and molecular abnormalities indicative of inflammation, treatments using certain anti-inflammatory agents have yielded limited results [60,119,123,125,126]. It is important to note that while dysferlin deficiency compromises plasma membrane integrity and disrupts intracellular equilibrium, other elements, such as specific calcium channels, might also be at play in the inflammation associated with dysferlinopathy [155]. The presence of non-selective Ca^2+^-permeable channels further complicate the inflammatory response tied to dysferlinopathy.

## 4. *DYSF* Mutations: Genotype–Phenotype Correlations and the Role of Dysferlin Partner Proteins

Despite extensive research, the phenotypic diversity of dysferlinopathies, including the lack of clear genotype–phenotype correlations, remains a challenge. Diversity within familial cases suggests external factors may play a role, including the rarity of the disease, absence of true-to-human disease animal models, and inherent clinical heterogeneity complicate investigations [7,22,30,42,60,156,157]. Also, as discussed in the previous section, our limited understanding of dysferlin’s function and its position in the overall protein network further complicates our comprehension of the disease. In addition, the compound heterozygosity prevalent in patients adds another layer of difficulty in interpreting individual mutations.

### 4.1. Genotype–Phenotype Correlations in Dysferlinopathies

Several studies have so far shown inconsistent genotype–phenotype correlations in dysferlinopathies, even within families [7,30,42,60,156,157]. A study of 12 patients from five closely related families in the Republic of Dagestan highlighted this inconsistency, with varied phenotypes despite identical mutations [42]. Similarly, diverse phenotypes within siblings have been recorded in Japanese, Canadian, and Italian families, regardless of identical gene mutations [76,157,158,159,160,161].

Recent advancements in genome analysis and larger sample sizes have begun to illuminate some patterns of the disease’s clinical manifestations, specific mutations, and potential modifier genes [22,23,162]. For instance, an international study of 69 patients suggested missense mutations in the *DYSF* gene correlate with higher CK levels and a more severe disease course than nonsense and frameshift mutations; however, a French study of 40 patients contradicted these findings [34,56].

Despite the complex landscape of dysferlinopathies, certain mutations are consistently associated with specific phenotypes, underscoring the need for larger cohort studies to understand the intricate pathobiology of this spectrum of diseases [7,42,48,51,157,159,163] (Table 1). For instance, a recent South Korean study with over a hundred patients has revealed that homozygous nonsense mutations, frameshift mutations, and splicing mutations result in a significantly earlier onset of the disease compared to patients with missense mutations [163]. Additionally, splicing mutations may lead to increased inflammation in muscles, although this is not a universal experience for all patients [7,67]. The interpretation of pathogenic variants, particularly missense mutations, is complicated by their potential effects on splicing, which can obscure correlations between mutation types and clinical features. For instance, among missense variants, *DYSF* G3370T and G3510A mutations are associated with variable impacts. The G3370T mutation is linked with a milder form of MM, featuring a later disease onset, while the G3510A mutation correlates with a more severe form of the disorder, characterized by earlier onset and higher serum CK levels [158,164].

Population-specific founder mutations identified in diverse populations reflect the genetic heterogeneity of dysferlinopathies [76,157,159,160,161] (Table 2). The most extensive study to date on genotype–phenotype correlations in dysferlinopathy patients, conducted in Japan in 2020, pointed out the need for large cohort studies to unravel the relations between specific nosologies, disease severity, and age of onset [48]. This study also suggested, for the first time, a potential mutational hotspot in the inner DysF domain of the dysferlin protein.

### 4.2. The Impacts of Pathogenic DYSF Mutations

Our understanding of the impacts of pathogenic mutations in the *DYSF* gene remains incomplete. Many *DYSF* mutations induce mRNA instability and degradation through the nonsense-mediated mRNA decay (NMD) process [177]. Other mutations result in the production of misfolded or unfolded proteins that may form aggregates [69]. The misfolded dysferlins are primarily degraded through the endoplasmic reticulum-associated degradation (ERAD) system and autophagy [179]. ERAD clears both native and mutant non-aggregated dysferlin, whereas autophagy eliminates aggregated mutated protein. In silico analysis reveals a bipartite nuclear localization sequence (NLS) near the dysferlin N-terminus [20]. Although endogenous dysferlin is typically not present in the nucleus, this NLS may be significant for truncated proteins lacking trans-membrane domains, potentially directing them to the nucleus. Thoroughly characterizing *DYSF* sequence variations is vital for developing effective therapeutic strategies [35].

### 4.3. The Role of Disease Modifiers

Proteins that partner with dysferlin have been reported to have modifier roles in many cases with dysferlinopathies. It has been suggested that the severity of dysferlinopathy and elevated levels of annexins A1 and A2 could have an association in effect, possibly due to a compensatory response to muscle damage [140]. Similarly, genetic variations in annexin genes could also affect the severity and progression of dysferlinopathies. Evidence shows that the 5979insA mutation in the *DYSF* gene results in decreased calpain-3 levels [142,161]. Similarly, *CAPN3* mutation can reduce dysferlin expression, highlighting their interaction in muscle fibers. *CAPN3* gene mutations may contribute to dysferlinopathy symptoms, and dysferlin may act as a modifier gene in calpainopathies [143]. Additionally, calpain-3 and dysferlin interaction involves AHNAK, which acts as a substrate for calpain-3, and contributes to calcium regulation within the muscle cells [180]. *AHNAK* mutations could influence both LGMD2A and dysferlinopathies. It would not be surprising at all if other proteins in the muscle cells also have modifier roles in dysferlinopathies. Deeper investigations on this, especially with a focus on the dysferlin’s partner proteins, with comprehensive functional analyses will likely identify modifiers of dysferlinopathies and their precise modifying role to explain the clinical heterogeneity of dysferlinopathies.

## 5. Animal Models for Studying Dysferlinopathies

The availability of animal models is critical for understanding the pathobiology of dysferlinopathies. Although there are some pathological differences between humans and these models, they have been instrumental in elucidating the causal gene relationships and the cellular and molecular mechanisms underlying dysferlinopathies. These models are particularly important for therapeutic development, having facilitated numerous preclinical studies that investigate therapeutic strategies, from proof-of-principle studies, dosage optimization, efficacy assessments, to safety evaluations and extended preclinical trials. Over the past decade, the research community has emphasized the need for comprehensive natural life history data from both patients and mouse models. Mice are most used in these studies because they are genetically similar to humans and can be easily manipulated genetically to mimic human diseases. Additionally, mice have a shorter lifespan and reproduce quickly, allowing researchers to study multiple generations in a relatively short period.

Several dysferlin-deficient mouse models are available for research, each created through various methods, e.g., exon suppression, retrotransposon insertion into an intron, point mutations from patients, and incorporation of partial or complete human dysferlin transgenes. Table 3 provides a detailed overview of the main disease characteristics of the most commonly used mouse models in preclinical research for dysferlinopathies.

The BLA/J mouse model, a hybrid of the A/J mouse on a C57BL/6 (B6) background, is extensively used in dysferlinopathy studies [181,182]. The first dystrophic signs, including centrally nucleated fibers and inflammation, appear at 2 months, with the psoas, quadriceps femoris, and TA muscles mainly affected [182]. Unlike the original A/J mice, BLA/J mice are not compromised by the C5 complement component deficiency, making them less susceptible to infections [182,183]. This model is particularly suited for therapeutic studies in gene therapy, stem cell therapy, and drug treatments focused on mitigating inflammation and muscle degeneration.

Preceding the BLA/J mice, the original A/J mice, an inbred strain, carry a ~6 kb retrotransposon in the intron 4 of dysferlin [184]. This disrupts *Dysf* gene splicing and eliminates dysferlin protein expression. These mice exhibit a phenotype similar to *Dysf^−/−^*, with onset at 4–5 months. The model also displays abdominal and proximal muscle deterioration, with only mild distal phenotypes. Its value lies in gene editing research, particularly for techniques like CRISPR/Cas9, aiming to correct the gene mutation and restore protein expression. It could also be useful in translational research for cell therapy and small molecule drug development targeting dysferlinopathies.

The SJL/J mouse model, with a splice-site mutation resulting in a 171 bp exon 45 deletion within *Dysf*, starts showing muscle weakness as early as 3 weeks, progressing to a more severe pathological state by 6 months [185]. This model exhibits a 15% residual dysferlin expression [186]. This model is useful for therapeutic studies focused on small molecule drugs that aim to increase dysferlin expression and gene therapy approaches to correct the splice-site mutation. It also offers a platform for antisense-based exon skipping therapies targeting exon 45.

*Dysf^−/−^* homozygous mice, engineered with a neomycin-resistant gene replacing the final three exons of the *DYSF* gene, show pathological symptoms, including muscle fiber degeneration and regeneration with central nuclei, starting at 2 months [184]. Initially, muscles like the quadriceps femoris are affected, with distal muscles showing pathology at later stages, usually at ages 5–6 months. These mice develop human-like pathological characteristics over time, e.g., necrotic fibers, phagocytosis, hypertrophy, splitting fibers, and fat accumulation. These mice could be key in studies evaluating the efficacy of small-molecule drugs and gene therapies targeting the dysferlin gene.

Another mouse model, the MMex38 mice, was developed to mimic missense mutant dysferlinopathy [187]. This model was created by introducing the murine mutation *Dysf* c.4079T > C in exon 38, resulting in the p.Leu1360Pro mutation, analogous to the human LGMDR2-causing *DYSF* c.4022T > C (p.Leu1341Pro) mutation. Notably, these mice do not exhibit symptoms at birth but start showing signs of muscular dystrophy, e.g., necrosis, regeneration of muscle fibers, fiber splitting, and fibrosis, from early adulthood, around 12 weeks. This progression, including increased fatty fibrosis by 60 weeks, closely resembles human disease progression. The MMex38 model is invaluable for exploring gene replacement and exon skipping therapies, providing a robust platform for studying dysferlinopathy treatments and their effects on disease progression.

Additionally, the B6.Cg-Tg(Ckm-DYSF)3Kcam/J model, developed by Kevin Campbell’s group at the University of Iowa, is a transgenic strain created by inserting the human dysferlin cDNA with SV40 polyadenylation signals downstream of the mouse *Ckm* (creatine kinase, muscle; MCK) promoter (https://www.jax.org/strain/014146; accessed on 13 October 2023). This model is especially beneficial for studying dysferlin’s role in muscular function and potential therapeutic applications targeting dysferlin in muscle diseases.

Additional models incorporate healthy human *DYSF* transgenes or patient-specific point mutations, e.g., c.4079T > C in exon 38, mirroring human variants like c.4022T > C (p.Leu1341Pro) [188]. Another notable model includes a mutation (c.3477C > A in exon 32) leading to a premature stop codon and loss of function [189]. These models are critical for studies evaluating the impact of specific gene mutations on dysferlin function and the success of gene editing approaches in rectifying these mutations.

It is crucial to note that while mice are predominantly utilized in dysferlinopathy studies, the progression of drugs to advanced clinical trial stages often necessitates large animal studies for a more accurate representation of human systems. Unfortunately, the availability of large animal models for the study of dysferlinopathies is currently lacking.

## 6. Therapeutic Approaches for Dysferlinopathies

Despite decades of extensive research, an effective and definitive cure for dysferlinopathy remains elusive; however, a variety of therapeutic strategies have been explored to mitigate the symptoms and address the genetic roots of this condition [35]. Table 4 provides a summary of the various therapeutic strategies for dysferlinopathies, highlighting their mechanisms, benefits, drawbacks, current status, and their suitability for different patient needs.

### 6.1. Symptomatic Treatments

Symptomatic treatments in dysferlinopathies focus on improving patient mobility and quality of life. This includes the use of walking aids, orthoses, and ankle–foot orthoses to compensate for muscle weakness. Physiotherapy and occupational therapy play vital roles in maintaining muscle function and flexibility [190]. Ankle–foot orthoses are also commonly employed. However, it is crucial to recognize that these interventions, while beneficial, are temporary and do not address the root cause of dysferlinopathy.

Experimental treatments, e.g., ezetimibe, are being explored to counter secondary effects like abnormal fat accumulation in muscles, a consequence of dysferlin deficiency [191]. Studies indicating its effectiveness in reducing fat deposition in dysferlin-deficient mice suggest a potential avenue for symptomatic relief.

Despite the known benefits of glucocorticoids in various inflammatory diseases and myopathies, including in many other types of muscular dystrophies, their use in dysferlinopathy has led to adverse effects, notably the loss of muscle strength and increased fatigability [192,193]. Clinical evidence has shown an accelerated decline in muscle strength in dysferlinopathy patients treated with glucocorticoids, e.g., deflazacort, compared to the natural progression of the disease [193]. Further studies have reinforced these findings, where high-dose steroid treatment in dysferlinopathy patients correlated with a faster disease progression and reduced muscle strength and mobility [54]. The exact mechanisms behind these adverse effects are not fully understood, but it is hypothesized that glucocorticoid-induced fat accumulation, insulin resistance, and muscle atrophy might contribute to the deterioration of muscle function in dysferlin-deficient individuals.

### 6.2. Pharmacological Approaches

Pharmacological treatments for dysferlinopathies offer diverse and innovative strategies targeting muscle repair, inflammation, and oxidative stress. These include proteasome inhibitors, Galectin-1 treatment, antioxidants, e.g., N-Acetylcysteine (NAC), and novel agents, e.g., halofuginone, each providing unique therapeutic benefits and illustrating the complexity of muscular dystrophy interventions.

#### 6.2.1. Proteasome Inhibitors

Dysferlin deficiency has been reported to cause an accumulation of TSP-1, a protein that exacerbates muscle inflammation and fibrosis [194]. TSP-1 is normally degraded by the proteasomes, but in dysferlinopathies, the proteasomes are overwhelmed by the excess of TSP-1 and other proteins. Therefore, by inhibiting the proteasomes, the levels of TSP-1 and other inflammatory and fibrotic factors may be reduced, potentially alleviating some of the symptoms of dysferlinopathies [195]. However, proteasome inhibitors do not directly affect the expression or function of dysferlin. Therefore, proteasome inhibitors may not be able to fully treat dysferlinopathies. It was shown that concomitant administration of proteosome inhibitors, e.g., oprozomib and ixazomib, with vitamin D3 has been observed to augment dysferlin expression in patient-derived myoblasts harboring exon 44 mutations [195]. This inhibition correlates with an upregulation of dysferlin and myogenin expression, although their impact on post-injury muscle membrane repair remains inconclusive.

#### 6.2.2. Galectin-1 Treatment

Galectin-1, a soluble carbohydrate-binding protein, has been found to enhance the myogenic potential and membrane repair capabilities in dysferlin-deficient models. This suggests galectin-1’s potential as a therapeutic agent for dysferlinopathies. Galectin-1 promotes myogenic maturation, as evidenced by improved myotube alignment, migration, and membrane repair capacity [196]. Recombinant galectin-1 (rHsGal-1) treatment leads to better myotube alignment, migration, and membrane repair, and even short-term treatment can enhance membrane repair independently of myogenic maturation [197]. Galectin-1’s carbohydrate recognition domain is crucial for this effect, facilitating calcium-independent membrane repair.

#### 6.2.3. Blockade of Hemichannels

The blockade of hemichannels represents a novel approach in dysferlinopathy management [198]. Dysferlin-deficient myoblasts, which often show a tendency towards fat accumulation rather than muscle formation, can be redirected towards normal muscle cell development using boldine, a connexin hemichannels blocker. This treatment potentially normalizes the differentiation of myoblasts and muscle features in BLA/J mice. Long-term boldine treatment has shown improvement in motor activity and muscle features.

#### 6.2.4. N-Acetylcysteine (NAC)

NAC is an acetylated cysteine residue that can directly scavenge reactive oxygen species [199]. NAC has been found to have an effect in reducing oxidative stress and improving muscle strength in BLA/J mice [200]. This is significant considering the oxidative stress observed in dysferlinopathy patients. Treatment with NAC significantly reduced oxidative markers in muscle tissues and improved muscle strength and resistance to fatigue, underscoring its potential in managing oxidative damage associated with dysferlinopathies.

#### 6.2.5. Diltiazem

Diltiazem, a calcium ion channel blocker, improves the contractile properties of skeletal muscle in BLA/J mice but does not appear to reduce contraction-induced muscle damage [201]. While it boosts muscle performance after eccentric contractions, it does not significantly reduce delayed-onset muscle damage.

#### 6.2.6. Halofuginone

Halofuginone enhances membrane repair and synaptotagmin-7 (Syt-7) levels in muscle cells of dysferlin-null mice [202]. Known for its anti-fibrosis effects, halofuginone has shown promising results in post-injury membrane resealing [203]. It reduces the percentage of membrane-ruptured myotubes in dysferlin-null mice and increases lysosome scattering, indicating elevated lysosomal exocytotic activity. This activity is linked to the spatial- and age-dependent expression of Syt-7, suggesting its compensatory role in the absence of dysferlin. Halofuginone does not impact key proteins in the patch–repair complex but enhances Syt-7 levels, suggesting a novel role in membrane-resealing events.

### 6.3. Molecular and Genetic Therapies

#### 6.3.1. Gene Replacement and Gene Editing Therapies

Gene therapy presents a potential treatment avenue for dysferlinopathies, but its development faces significant challenges [23]. The enormous size of DYSF, difficulties in targeting degenerating muscles, and possible immune responses to viral vectors or the dysferlin protein itself are major hurdles. However, several recent studies have explored various approaches to circumvent these obstacles.

Explorations into direct dysferlin replacement using both full-length and truncated forms have been undertaken [204,205]. The use of a dual AAV vector system successfully delivered the dysferlin gene to muscle tissues, achieving sustained expression and reduced muscle damage [206]. Similarly, delivering a truncated dysferlin molecule via a single AAV vector has been tested [204]. In another study, naked plasmid DNA was injected vascularly to deliver therapeutic protein to hind limb muscles in BLA/J mice, which resulted in the rescue of dysferlin in the muscle fibers [207]. In addition, a novel strategy involved non-muscle targeted gene therapy, which showed promise by injecting a liver-targeting AAV vector expressing human acid sphingomyelinase (ASM), a key protein in muscle membrane repair [208]. This approach improved muscle function and reduced degeneration in a dysferlin deficiency mouse model.

Cutting-edge gene-editing technologies, e.g., CRISPR/Cas9 and base or prime editing, offer precise correction methods for the DYSF gene mutations [209,210]. Base editing uses a modified CRISPR-Cas9 system that can change one base to another, using a deaminase enzyme that converts the target base to an intermediate base, which is then recognized and fixed by the cellular DNA repair machinery [209]. Prime editing uses a longer guide RNA, called a prime editing guide RNA (pegRNA), and a fusion protein of a Cas9 nickase and a reverse transcriptase, which can copy the desired edit from the pegRNA to the target DNA strand [210]. Base editing and prime editing are precise genomic targeting techniques with potential for correcting dysferlinopathies. However, they come with limitations. Base editing is restricted to four types of base conversions, i.e., C to T, T to C, A to G, G to A, potentially limiting its applicability to many pathological mutation in *DYSF*. In contrast, prime editing offers broader capabilities, enabling all 12 types of base conversions. Yet, it demands a more complex and longer pegRNA design and tends to have lower efficiency and higher off-target activity compared to base editing. Additionally, both methods face challenges in effectively delivering the edits to degenerating muscles, possibly necessitating the use of viral vectors, nanoparticles, or ex vivo techniques.

Ongoing research is focused on optimizing these gene therapy techniques to enhance their safety, efficacy, and delivery. While promising, these approaches require further development and clinical testing to establish their feasibility and effectiveness in treating dysferlinopathies.

#### 6.3.2. Antisense-Mediated Exon Skipping

Antisense-mediated exon skipping, utilizing antisense oligonucleotides (ASOs), is a cutting-edge therapy for dysferlinopathies [211,212]. This method targets and skips specific exons in a gene, altering mRNA splicing to bypass mutations and restore the gene’s correct reading frame. ASOs are delivered through various means, including conjugation with peptides, lipids, or antibodies, or via stable nucleic acid lipid particles and exosome-based systems [211].

While this approach has been extensively applied in Duchenne muscular dystrophy (DMD), its use in dysferlinopathies slightly differs as it targets exons containing mutations [213,214,215]. Four exon-skipping drugs have so far received FDA approval for clinical use in DMD [216,217,218,219]. These drugs aim to skip exons that flank out-of-frame deletions in the dystrophin gene. But, for dysferlinopathies, the targeted exons typically contain the mutation. Successful skipping of these exons can result in a truncated yet partially functional dysferlin protein. Dysferlin’s modular nature allows it to retain some functionality despite missing domains, a hypothesis supported by patients exhibiting milder symptoms due to natural exon skipping [164]. Several studies have demonstrated the efficacy of this technique in increasing dysferlin expression and enhancing membrane repair, suggesting its potential in treating various dysferlin gene mutations [151,220]. Studies from our lab suggests restored human fibroblast plasma membrane resealing in response to skipping exons 26–27 or 28–29 [221]. Malcher and colleagues showed that exons 37 and 38 can also be skipped in MMex38 mice to restore resealing [187]. These studies indicate the potential of exon skipping in addressing various mutations within the dysferlin gene, highlighting its versatility and specificity as a therapeutic approach.

#### 6.3.3. Membrane Repair and Stabilization

With an aim to improve membrane repair in dysferlin-deficient muscle cells, several therapeutic approaches are under investigation, including the use of recombinant human MG53 (rhMG53) protein and modified steroids [222]. RhMG53 has been shown to enhance membrane repair processes in B6.129-Dysf^tm1Kcam^/J mice. This treatment increases the integrity of the sarcolemmal membrane in muscle fibers, both in isolation and in whole muscles, in a calcium-independent manner. Additionally, pre-treatment with rhMG53 before strenuous exercise can reduce muscle damage and prevent the leakage of intracellular enzymes and proteins. These findings indicate that short-term rhMG53 treatment can effectively improve sarcolemmal membrane integrity in dysferlin-deficient muscles.

### 6.4. Cell-Based and Tissue Engineering Approaches

Cell-based and tissue engineering approaches are playing an increasingly pivotal role in advancing our understanding and treatment of dysferlinopathies. Three-dimensional skeletal muscle tissue models create a controlled environment for in-depth disease study and therapeutic testing, providing valuable insights at the tissue level. Additionally, patient-derived induced pluripotent stem cells (iPSCs) are instrumental for phenotypic drug screening, offering a tailored approach to explore disease mechanisms and evaluate potential treatments [223].

Myoblast transplantation, involving the injection of dysferlin-positive cells into muscle tissue, has shown promise in dysferlinopathy research. For instance, Leriche-Guérin et al. (2002) demonstrated that transplanting myoblasts in SJL dysferlin-deficient mice led to a significant restoration of dysferlin-positive fibers [224,225]. This technique, however, faces significant challenges, e.g., limited myoblast migration, necessitating multiple injections per muscle and potential immunological issues. A notable advancement in this area is the use of the Sleeping Beauty (SB) system for stable gene transfer. The SB system, composed of a plasmid carrying the full-length DYSF flanked by transposon sequences and the transposase SB100X coding sequence, was employed to correct dysferlin-deficient mouse myoblasts and successfully engraft them in immunodeficient BLA/J (SCID BLA/J) mice, yielding positive results [226].

In addition, stem cell therapies, especially those using mesenchymal stem cells (MSCs) and iPSCs, hold theoretical promise in treating dysferlinopathies. However, current limitations in stem cell technology, e.g., the challenge of ensuring stem cells differentiate into specific cell types, have impeded the development of effective treatments for dysferlinopathies. It is important to approach unverified stem cell therapies with caution due to potential risks and a lack of scientific support. Ongoing research, including a study with patient-derived iPSCs showing the effectiveness of nocodazole in increasing dysferlin levels in cells, indicates promising directions for future dysferlinopathy treatments [223].

### 6.5. Dietary and Metabolic Approaches

#### 6.5.1. Ketogenic Diet

The ketogenic diet has shown promise in improving dysferlinopathy symptoms in animal models by enhancing mitochondrial function. This diet’s potential benefits include improved muscle function, regeneration, anti-inflammatory effects, and neuroprotection. In a mouse model, the diet improved muscle strength, reduced inflammation, and boosted muscle repair gene expression [227]. However, its efficacy and safety in humans remain to be conclusively studied. Future clinical trials should assess the long-term impacts and risks of this dietary intervention, including its ideal macronutrient composition and duration. Given the potential side effects on organs like the liver and kidneys, medical consultation and regular monitoring are essential before and during a ketogenic diet.

#### 6.5.2. AMPK Complex Activation

AMP-activated protein kinase (AMPK) complex activation plays a crucial role in sarcolemmal repair in dysferlinopathy, emphasizing the importance of cellular metabolism in managing the disease. Research involving recombinant proteins, affinity purification, and liquid chromatography–tandem mass spectrometry (LC-MS/MS) revealed that AMPKγ1 binds to a specific region of dysferlin [228]. Ex vivo experiments showed that the AMPK complex is vital for repairing sarcolemmal damage in skeletal muscle fibers, with its accumulation during injury being calcium dependent and regulated by dysferlin. The phosphorylation of AMPKα was found to be essential for plasma membrane repair. Treatment with the AMPK activator metformin improved muscle phenotype in zebrafish and mouse models of dysferlin deficiency, highlighting the AMPK complex as a potential therapeutic target for dysferlinopathy.

### 6.6. Clinical Trials for Dysferlinopathies

Clinical trials for dysferlinopathies have evolved significantly over the past two decades, yielding diverse outcomes that have substantially enriched our understanding and treatment approaches.

The Ludwig-Maximilians University of Munich conducted a pivotal phase 2/3 study (NCT00527228), critically evaluating Deflazacort, a corticosteroid, in treating dysferlinopathy. This double-blind trial, including a placebo group, concluded that Deflazacort was ineffective in enhancing muscle strength and had adverse side effects. This finding was crucial in establishing that steroids, commonly used for conditions like polymyositis, are not suitable for dysferlinopathy.

In a novel approach, the University Hospital in Basel, Switzerland, embarked on a Phase 1 trial (NCT01863004), investigating proteasomal inhibition in patients with missense-mutated dysferlin. Inspired by successful in vitro studies [229,230], this trial hypothesized that proteasomal inhibition could slow the degradation of dysferlin, crucial for muscle membrane repair, thus increasing its functional presence in patients.

Sarepta Therapeutics in the United States conducted an important Phase I trial (NCT02710500), testing a novel dual-vector AAV gene therapy, SRP-6004 (rAAVrh.74.MHCK7.DYSF.DV), through intramuscular injections into the extensor digitorum brevis muscle. Following promising preliminary experiments in mice and non-human primates, which showed restored dysferlin expression and function, the clinical trial’s unpublished results are highly anticipated [225,231]. Additionally, Sarepta’s ongoing trial (NCT05906251) is assessing the intravenous administration of SRP-6004 to adult ambulatory patients, further highlighting the potential of gene therapy in treating dysferlinopathies.

These studies collectively underscore the complexities of managing this muscular dystrophy and point towards a need for precision medicine, efficient delivery systems, and patient-centric outcomes in future trials. The dysferlinopathy community, including patients, caregivers, and researchers, keenly anticipates the results of these trials, particularly for their potential in transforming treatment landscapes. The challenges in defining reliable surrogate endpoints and the importance of advanced imaging, expanded patient populations, and international collaboration further emphasize the need for a multifaceted approach in upcoming trials.

## 7. Conclusions

Dysferlinopathies represent a complex diagnostic and therapeutic challenge, featured by their clinical heterogeneity and the intricate molecular mechanisms underlying their pathogenesis. While advancements in genetic and molecular technologies have deepened our understanding, they have also revealed the vast complexity of the disease. Current therapeutic strategies, ranging from symptomatic support to innovative molecular therapies, offer potential yet underscore the need for individualized treatment approaches. The development of animal models has provided invaluable insights, although differences between model systems and human disease necessitate cautious interpretation. Notably, the exploration of pharmacological and genetic therapies has shown promise, but further research is required to overcome delivery challenges and optimize efficacy. Clinical trials remain to be the cornerstone for evaluating these emerging therapies, with patient-centric outcomes being paramount. The future of dysferlinopathy treatment lies in the synergy of comprehensive research, innovative therapeutic strategies, and collaborative efforts that prioritize patient well-being and quality of life. As we stand on the cusp of significant breakthroughs, it is imperative to maintain a steadfast commitment to translating scientific discoveries into tangible clinical benefits for patients grappling with the realities of dysferlinopathies.

## Figures and Tables

**Figure 1 biomolecules-14-00256-f001:**
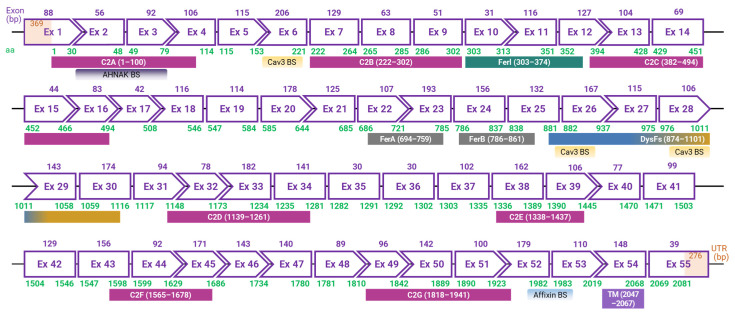
Exon map and phasing of human dysferlin based on the canonical transcript. Depicted is the exon (Ex) configuration of the human dysferlin gene, comprising 55 exons. Rectangles represent in-frame exons; chevron sides represent exon junctions that occur within the codons at the end of the phased exons. Above each exon, the length is annotated in base pairs (bp, purple). Beneath each, the encoded amino acid (aa, green) range is denoted. Also, the domains and binding sites (BS) for Cav3, AHNAK, and Affixin are delineated, accompanied by the respective aa range constituting each domain. The untranslated regions (UTR) at the 5′ and 3′ ends are shown in pale orange, with their base pair counts (bp, orange) inscribed within. Note: scale is indicative, not precise.

**Figure 2 biomolecules-14-00256-f002:**
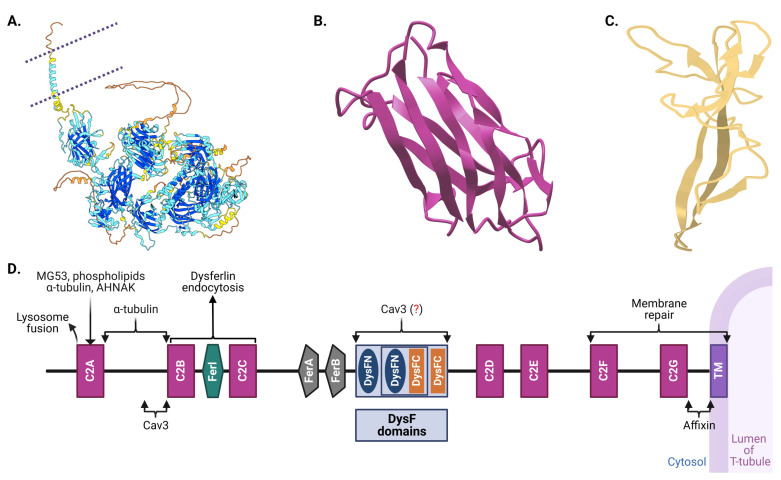
Structural overview of dysferlin protein. (**A**) Dysferlin, a type II transmembrane protein, is associated with various proteins important for membrane trafficking and repair. Shown is a cartoon structure view of the human dysferlin protein, predicted using the Swiss-Model protein structure prediction tool. The template for this prediction was the AlphaFold DB structure A6QQP7.1 (DYSF_BOVIN; protein sequence identity; 93.56%, coverage: 100%) corresponding to the human dysferlin sequence (GenBank protein AAC63519.1). The quality of this model was validated with Ramachandran plotting, where 92.97% of residues were found in favored regions, and only 2.26% of residues fell in outlier regions. The plasma membrane is demarcated by purple dotted lines. C2A and DysF domain sub-structures in (**A**) are enlarged in (**B**,**C**), respectively. (**D**) A simplified linear structure of human dysferlin, enumerating distinct domain functionalities [91,99]. Shown are the seven calcium-binding C2, Ferl, FerA and FerB, DysFNs and DysFCs, and transmembrane domains. Binding sites for important dysferlin partner proteins are shown as well. Dysferlin’s C2 domains have multifaceted roles: they facilitate targeting to the transverse tubules, facilitate membrane repair, modulate the amplitude of the Ca^2+^ transient, influence the Ca^2+^ transient in response to osmotic shock injury, and mitigate Ca^2+^ surges subsequent to such shocks [100,101]. The fer domain is known for its calcium-dependent membrane interactions 101]. The precise role of the DysF domains remains under investigation, though its mutations are frequently implicated in muscular dystrophies [89,90,91,92,93,94,95,96,97]. Note: the figures are schematic and not to scale; color coding is used for illustrative purposes only. In (D), proteins potentially interacting with dysferlin are marked with a question mark in parentheses, indicating theoretical binding regions. (Created with BioRender: US266PDI0R).

**Figure 3 biomolecules-14-00256-f003:**
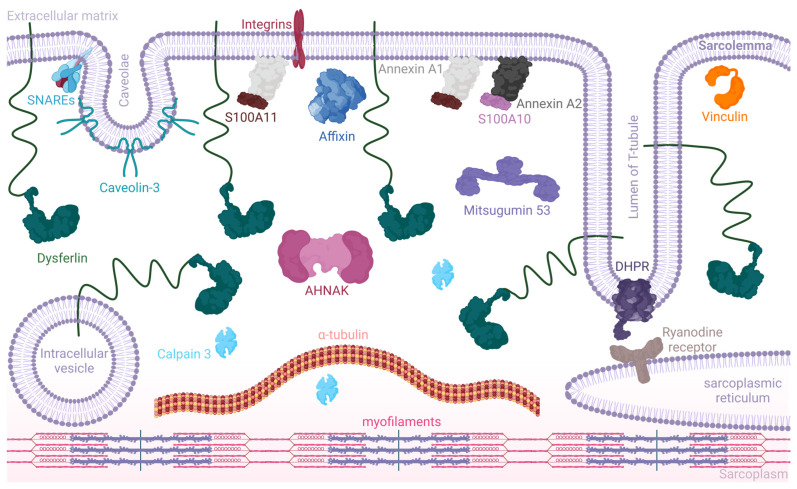
Dysferlin and its major partner proteins in the sarcolemma (simplified). This schematic illustrates dysferlin anchored to the sarcolemmal membrane, T-tubule vesicle, and intracellular vesicles. It associates with a variety of proteins, including but not limited to AHNAK, essential for membrane organization and repair; SNARE proteins, which play a pivotal role in vesicle fusion; affixin, involved in linking the actin cytoskeleton to the membrane; annexin A1 complexed with S100A10 and annexin A2 complexed with S100A11, critical for calcium-dependent membrane repair processes; calpain 3, a protease involved in muscle remodeling; caveolin 3, associated with membrane curvature and muscle fiber repair; the dihydropyridine receptor (DHPR), essential for calcium channel regulation; mitsugumin 53, involved in membrane repair; as well as α-tubulin and vinculin, which are key components of the cytoskeleton. Dysferlin’s interaction with these proteins is crucial for a range of functions from vesicle trafficking and membrane fusion to muscle contraction and calcium signaling. Note: the figure is not to scale, and color coding is used for illustrative purposes to enhance visual distinction (created with BioRender: NB266PDFC4).

**Table 1 biomolecules-14-00256-t001:** Clinical Features across Dysferlinopathies Disease Spectrum. This table summarizes the main features and differences of the three major and two less frequently observed phenotypes within the spectrum of dysferlinopathies [22,23,29,31,32,34,43,44,45,46,47,48,49,50,51,52,53,54,55,56,57].

	Dysferlinopathies Phenotypes				
Features	Limb-Girdle Muscular Dystrophy Recessive Type 2	Miyoshi Myopathy	Distal Myopathy with Anterior Tibial Onset	Dysferlin-Deficient Proximo-Distal Phenotype	Asymptomatic HyperCKemia
Age of onset	Late teen to thirties	Late teen to thirties	Early adulthood	Variable, usually early twenties to early thirties	Variable, usually >50 years
Type of muscular dystrophy	Proximal	Distal	Distal (anterior tibial)	Proximal and distal combined	–
Atrophy	Present	Present	Present	Present	No
Initial symptoms	Weakness and atrophy of the proximal muscles, especially the gluteus maximus (buttock) and quadriceps (thigh) muscles	Weakness and atrophy of the distal muscles, especially the gastrocnemius (calf) and soleus (lower leg) muscles	Weakness and atrophy of the distal muscles, especially the anterior tibialis (shin) and extensor digitorum longus (toe extensor) muscles	Weakness and atrophy of both proximal and distal muscles, with variable distribution and severity	Only elevated serum CK levels with no or insignificant muscle weakness or other symptoms
Progression	Slowly progressive, spreading to other proximal muscles and eventually affecting distal muscles; usually symmetrical	Slowly progressive (may be faster than LGMDR2), spreading to other distal muscles and eventually affecting proximal muscles; usually symmetrical	Slowly progressive (may be faster than LGMDR2 and MM), spreading to other distal muscles and sometimes affecting proximal muscles; usually symmetrical	Slowly (variable compared to LGMDR2, MM, and DMAT) progressive, affecting both proximal and distal muscles in a symmetrical or asymmetrical pattern; may have focal or regional involvement	Stable or fluctuating CK levels with no or insignificant muscle involvement
CK levels (times normal)	Very high (50–200)	Very high (50–200)	High (20–70)	High (10–50)	High (5–10)
Muscle biopsy findings	No/very low levels of dysferlin in muscle fibers; muscle damage with degeneration and regeneration of muscle fibers; inflammation with cells that invade the muscle tissue; scar tissue formation and fat deposits that replace muscle tissue; these changes can vary in severity and distribution among different types of dysferlinopathy				No/low levels of dysferlin in muscle fibers; mild or no changes in muscle fibers; no inflammation, scar tissue, or fat deposits
Cardiac involvement	Rare, 3–10% patients may develop cardiac dysfunction or arrhythmias; may require cardiac monitoring and treatment if present				
Respiratory involvement	Uncommon, 20–30% patients may develop respiratory impairment and/or sleep apnea; may require respiratory monitoring and treatment if present				
Life expectancy	Not significantly impacted unless cardiac/respiratory involvement is observed				Not affected, as there is no muscle involvement or other complications
Quality of life	Causes significant disability; may result in complete loss of ambulation; may impact psychosocial well-being; may require multidisciplinary care and support to cope with the challenges and improve the function				May not be affected; some patients may experience stress and anxiety due to elevated CK levels and potential future risk of muscle weakness

**Table 2 biomolecules-14-00256-t002:** Correlations and implications of select *DYSF* gene mutations. This details the associated phenotypes, impact on the protein, and potential pathobiological implications. It includes mutations based on various criteria: the first mutation discovered, prevalence in the population, unique or notable phenotypes, historical significance, atypical mechanisms of mutation, distinctive pathogenetic mechanisms, and specific patterns of inheritance, e.g., founder effects. Note that this compilation is not exhaustive but aims to highlight key mutations for their relevance and distinct characteristics.

Mutation	Associated Phenotype	Impact on Protein	Pathobiological Implications
c.573-574TG > AT (p.Val67Asp) [40,42,165]	MM, proximo-distal, LGMDR2	Affects calcium binding	Altered muscle contraction
c.1867C > T (p.Gln623Ter) [2]	MM	Nonsense mutation causing premature truncation	Diverse clinical manifestations
c.2372C > G (p.Pro791Arg) [20,67,163]	LGMDR2, mild distal myopathy (similar to MM), asymmetric hypertrophy, mild proximal muscle weakness	Unclear; probable protein instability and/or malfunctionality	Diverse clinical manifestations; founder effect observed in a Canadian Aboriginal population
c.1566C > G (p.Tyr522X) [48,162]	LGMDR2, MM (MM is more common)	Leads to mRNA instability	Earlier disease onset
c.2997G > T (p.Trp999Cys) [162,166,167,168,169]	LGMDR2, MM (LGMDR2 is more common)	Unclear; probable protein instability and/or malfunctionality	Late onset and a milder course of the disease
c.3373del (p.Glu1125LysfsX9) [49]	MM, LGMDR2 (sporadic)	Unclear; probable protein instability and/or malfunctionality	Diverse clinical manifestations
c.3946A > G (p.Ile1316Val) [2,6,170,171,172,173]	MM < LGMDR2, DMAT	Unclear; probable protein instability and/or malfunctionality	Diverse clinical manifestations
c.6135G > A (p.Trp2045X) [48,174]	MM	Unclear; probable protein instability and/or malfunctionality	Diverse clinical manifestations
c.1609G > A (p.Gly537Arg) [70,175]	MM	Unclear; probable protein instability and/or malfunctionality	May result in late onset milder manifestation
c.1927G > T (p.Asp643Tyr) [70]	LGMDR2	Unclear; probable nonfunctional protein	Associated with late onset, progressive fatigue, increased serum CK levels, and fatty infiltrations in the lower limb muscles
c.3497-33A > G [176]	LGMDR2	Intronic mutation resulting in the in-frame large deletion of exon 32, resulting in a significantly reduced production of the protein	May be associated with a milder manifestation
c.IVS12 + 7delAGTGCGTG (c.1180 + 7delAGTGCGTG) [177]	MM, proximo-distal phenotype, LGMDR2	Intronic mutation resulting in abnormal splicing	Diverse clinical manifestations; founder effect observed in a Portuguese population
c.2779delG (p.Ala927LeufsX21) [164]	MM, proximo-distal phenotype, LGMDR2, asymptomatic hyperCKemia, congenital phenotype	Unclear; frameshift mutation possibly leading to a truncated, likely nonfunctional protein	Diverse clinical manifestations; founder effect observed in Caucasian Jewish population
c.2875C > T (p.Arg959Trp) [178]	May result in a milder phenotype	Unclear; probable protein instability and/or malfunctionality	Diverse clinical manifestations; founder effect observed in an Italian population
c.3191G > A (p.Arg1064His) [6,32,33,144,162,166,179]	MM	Unclear; probable protein instability and/or malfunctionality	Associated with early onset and significantly higher CK levels
c.4989_4993delinsCCCC (p.Glu1663fs) [20,32,34,65,161,176,180]	May result in severer phenotypes	Complex mutation involving deletion and insertion, leading to a frameshift and truncated protein	Likely associated with severe manifestation of the disease; founder effect observed in Lebanese Jewish population
c.5156_5174 + 4dup 23-bp ins [65]	LGMDR2	Tandem duplication resulting from replication slippage, and was predicted to result in frameshift and premature termination	Diverse clinical manifestations
c.5174 + 5G > A [69]	LGMDR2	Intronic mutation resulting in abnormal splicing	Associated with elevated CK levels, presence of inflammatory process in histopathology
c.5492G > A (p.Gly1831Arg) [181]	LGMDR2, MM, extreme hypertrophy, asymptomatic hyperCKemia, cardiac arrhythmia	Unclear; possible altered mRNA splicing	Diverse clinical manifestations; founder effect observed in a Portuguese population
c.5713C > T (p.Arg1905X) [78]	MM, LGMDR2, DMAT, proximo-distal phenotype	Nonsense mutation leading to premature protein termination	Likely associated with severe phenotypes; founder effect observed in a Spanish population
c.6241C > T (p.Arg2081Cys) [2,6,32,47,72,135,143,182,183]	MM, LGMDR2	Unclear; probable protein instability and/or malfunctionality	Diverse clinical manifestations

**Table 3 biomolecules-14-00256-t003:** Overview of mouse models to study dysferlinopathies. This table outlines details of the strain, background, genetic makeup, and the clinico-pathological manifestations observed in each model [100,181,182,183,184,185,186,187,188,189].

Strain	Background	Genetic Makeup	Clinico-Pathological Manifestation
BLA/J (B6.A-*Dysf^prmd^*/GeneJ)	C57BL/6J	Spontaneous ETn retrotransposon insertion in intron 4; no dysferlin expression	Dystrophic features by 4–5 months; loss of muscle mass, lipid deposition by 8 months; slow progression; limb girdle, psoas, quad most affected.
A/J	Inbred A/J	Spontaneous ETn retrotransposon insertion in intron 4 resulting no expression of dysferlin protein	Similar to BLA/J, but initial proximal bias; rapid abdominal muscle wasting; mild cardiomyopathy at ~10 months; hearing loss; lung adenomas; C5 deficiency; susceptibility to infections.
SJL/J	Wild-derived Swiss mice	Splice site mutation in exon 45; ~15% normal dysferlin expression	Dystrophic features by 2–4 months; pronounced histopathology by 6–8 months; enhanced inflammation; faster progression; aggression; high lymphoma incidence; susceptibility to autoimmune diseases and infections.
129-*Dysf^tm1Kcam^*/J B6-*Dysf^tm1Kcam^*/J	129 C57BL/6J	Neomycin resistance gene replacement causing deletion of the last three coding exons or transmembrane domain	Dystrophic features by 2 months; pronounced histopathology by 8 months; psoas most affected; mild cardiomyopathy with fibrosis from 12 to 14 months, worsens with cardiac stress exercise
MMex38	129	Introduction of the missense c.4079T > C mutation in exon 38 of murine *Dysf*; this mutation is analogous to a clinically relevant mutation (p.Leu1341Pro) in human	Dystrophic features by 12 weeks of age; exhibits a progressive dystrophic pattern, amyloid formation, and defects in membrane repair

**Table 4 biomolecules-14-00256-t004:** Overview of therapeutic approaches for dysferlinopathies. This table lists various treatment strategies for dysferlinopathies, encompassing symptomatic treatments, pharmacological interventions, molecular and genetic therapies, cell-based and tissue engineering approaches, and dietary and metabolic methods. It summarizes the mechanisms of action, advantages, disadvantages, current translational status, and applicability of each approach, offering an insightful guide for understanding the multifaceted efforts in managing and potentially treating dysferlinopathy.

Therapeutic Approach	Description	Pros	Cons	Translational Status	Applicability
Symptomatic Treatments					
Physical aids (e.g., orthoses)	Aims to compensate for muscle weakness	Improves mobility and independence	Does not address underlying disease	Widely used in clinical practice	Suitable for most patients
Physiotherapy and occupational therapy	Helps maintain muscle function and flexibility	Reduces symptom progression	Requires ongoing commitment	Standard supportive care	Broad applicability
Experimental drugs (e.g., ezetimibe)	Counters abnormal fat accumulation	Targets secondary effects	Limited evidence in humans; potential side effects	Early research stages	Limited to specific symptoms
Corticosteroids (e.g., Deflazacort)	Anti-inflammatory and immunosuppressive	Beneficial in other muscular dystrophies	Adverse effects in dysferlinopathy, e.g., muscle weakness	Clinical evidence suggests accelerated decline in muscle strength	Not recommended for dysferlinopathy
Pharmacological Approaches					
Proteasome inhibitors	Reduce levels of TSP-1, decrease inflammation and fibrosis	Reduces inflammation and fibrosis	May not directly affect dysferlin expression	Preclinical and early clinical trials	Potentially applicable to a subset of patients
Galectin-1 treatment	Enhances myogenic potential and membrane repair	Improves muscle repair capabilities	Long-term effects unknown	Experimental stage	Limited by stage of muscle degeneration
N-acetylcysteine (NAC)	Scavenges reactive oxygen species	Reduces oxidative stress	Efficacy in humans not fully established	Preclinical and early clinical trials	Broad applicability for oxidative stress management
Diltiazem	Calcium ion channel blocker, improves muscle contractility	Improves muscle contractile properties	Does not reduce muscle damage	Preclinical evidence	Limited applicability; more research needed
Halofuginone	Enhances membrane repair, increases lysosomal exocytotic activity	Enhances membrane repair	Limited human data; potential side effects	Early clinical trials	Patients with severe membrane repair deficits
Molecular and Genetic Therapies					
Gene replacement/editing	Restores functional dysferlin protein	Potential for long-term treatment	Technical challenges; risk of immune response	Early clinical trials (gene therapy)	Limited to patients with specific mutations
Antisense-mediated exon skipping	Alters mRNA splicing to bypass mutations	Can restore functional protein	Efficacy varies; potential off-target effects	Early clinical trials	Specific mutations amenable to exon skipping
Cell-based and Tissue Engineering					
Myoblast transplantation	Restoration of dysferlin expression in muscles	Potential restoration of dysferlin expression	Limited migration, immunological issues	Experimental stage	Suitable for localized treatment
Stem Cell Therapies (MSCS, iPSCS)	Differentiation into muscle cells to replace damaged tissue	Theoretical potential for muscle regeneration	Technical challenges in differentiation and delivery	Preclinical stage	Future potential; currently experimental
Dietary and Metabolic Approaches					
Ketogenic diet	Enhances mitochondrial function and muscle repair	Improves mitochondrial function in models	Long-term human efficacy unknown	Dietary intervention; clinical trials needed	Broad applicability with medical supervision
AMPK complex activation	Vital for repairing sarcolemmal damage in skeletal muscle fibers	Important in muscle repair	Limited human data	Preclinical and early clinical trials	Potential for widespread use in muscle repair

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
