# Peer review of "The Dysferlinopathies Conundrum: Clinical Spectra, Disease Mechanism and Genetic Approaches for Treatments"

_biomolecules, 2024, doi:10.3390/biom14030256_

Round 1

Reviewer 1 Report

Comments and Suggestions for Authors

The manuscript by Anwar and Yokota summarizes a current understanding of dysferlinopathies by comprehensively covering important topics, such as biology/physiology of dysferlin as well as pathological feature of dysferlinopathies. Selection of topics is made in a way meeting the demands of both research and clinical fields. There are some expressions which blur the context or meaning. Please reconsider the following points.

(l. = line)

l. 201; “team” should be time.

l. 250; Definition of “ferlin family”, and if possible, a regular domain composition, should be provided.

l. 265, 266; Fig. 2C could be modified/colored to better depict what “nested structure” means. As to “gene duplication events”, an appropriate reference should be placed after this sentence.

l. 252; please check the sentence for grammar.

l. 374; “3.2.6”

l. 375-415; The information provided in Fig. 2 should be appropriately cited.

l. 399; “be involved in” should be correct.

l. 404-411; As an introduction of this section, the first sentence could be modified. The following contents should be described in a manner similar to that applied in lines 375-403. Then, lines 412-414 could be rewritten accordingly.

l. 427; Appropriate introduction for “myoferlin” should be provided. This modulation should be made in unison to the comment for line 250.

l. 474-475; It is not clear why only “missense” variants are mentioned as to their effects on splicing. Could it be more reasonable to start the sentence (in l. 476) as “For instance, among missense variants, …”?

l. 481-482/489-491 and Table2; For mutations summarized in Table 2, labeling corresponding to l. 481-482/489-491 is missing and it is impossible to understand which mutation is included in the table based on which criteria for inclusion. Table 2 should be modified and each mutation should be accompanied with an appropriate reference info.

l. 758-759; Combination of “address”, “therapies”, and “MG53…” seems illogical for “therapies” are used to improve the current status, and not equal to “MG53” or “steroids” (they are therapeutic reagents). Revising one or two expression(s) should be considered.

Fig. 2; For 2A-2C, domain structures should be properly labeled. For 2B and 2C, IDs of original data should be explained. For 2D, information about “function” and “interactor” should be depicted differently (current design is misleading due to both arrows and brackets being mixed).

Author Response

Response to Reviewer 1

Reviewer 1: The manuscript by Anwar and Yokota summarizes a current understanding of dysferlinopathies by comprehensively covering important topics, such as biology/physiology of dysferlin as well as pathological feature of dysferlinopathies. Selection of topics is made in a way meeting the demands of both research and clinical fields. There are some expressions which blur the context or meaning. Please reconsider the following points.

Author(s): We thank you for acknowledging the comprehensive coverage of dysferlinopathies in our manuscript and appreciate your constructive feedback.

Reviewer 1: l. 201; “team” should be time.

Author(s): Thank you for pointing out this typographical error. It has been corrected in the revised manuscript.

Reviewer 1: l. 250; Definition of “ferlin family”, and if possible, a regular domain composition, should be provided. l. 427; Appropriate introduction for “myoferlin” should be provided. This modulation should be made in unison to the comment for line 250.

Author(s): Your suggestion is well-received. We have expanded the discussion on ferlins and myoferlins to provide a more comprehensive understanding.

Reviewer 1: For Fig. 2A-2C, domain structures should be properly labeled. For 2B and 2C, IDs of original data should be explained. For 2D, information about “function” and “interactor” should be depicted differently (current design is misleading due to both arrows and brackets being mixed). Fig. 2C could be modified/colored to better depict what “nested structure” means. As to “gene duplication events”, an appropriate reference should be placed after this sentence. The information provided in Fig. 2 should be appropriately cited.

Author(s): We have revised Fig. 2A-2C to enhance clarity and accuracy, ensuring proper labeling and explanation of domain structures. We have revised the design in Figure 2D to distinctly differentiate between 'function' and 'interactor' information. For Figure 2C, we believe that an attempt to illustrate the 'nested structure' might inadvertently reduce clarity. Instead, we have focused on elucidating this concept in Figure 2D. Appropriate references to support the information presented in Figure 2 have been added, ensuring it is both comprehensive and informative. Additionally, references supporting the claim that the nested structure of the DysF domain results from gene duplication events have been cited.

Reviewer 1: l. 252; please check the sentence for grammar.

Author(s): The sentence in question has been reviewed and has been edited for grammatical accuracy.

Reviewer 1: l. 374; “3.2.6”

Author(s): The typographical error has been rectified in the revised manuscript.

Reviewer 1: l. 399; “be involved in” should be correct.

Author(s): We have revised this sentence for better clarity and coherence.

Reviewer 1: l. 404-411; As an introduction of this section, the first sentence could be modified. The following contents should be described in a manner similar to that applied in lines 375-403. Then, lines 412-414 could be rewritten accordingly.

Author(s): We have updated this section to reflect a consistent style and clarity as per your suggestion.

Reviewer 1: l. 474-475; It is not clear why only “missense” variants are mentioned as to their effects on splicing. Could it be more reasonable to start the sentence (in l. 476) as “For instance, among missense variants, …”?

Author(s): This part has been updated to clarify the specific focus on missense variants and their effects.

Reviewer 1: l. 481-482/489-491 and Table2; For mutations summarized in Table 2, labeling corresponding to l. 481-482/489-491 is missing and it is impossible to understand which mutation is included in the table based on which criteria for inclusion. Table 2 should be modified and each mutation should be accompanied with an appropriate reference info.

Author(s): This section has been updated to clarify the specific focus on missense variants and their effects. To ensure the table offers valuable insights, we have carefully selected mutations based on criteria such as the first mutation discovered, high population frequency, distinctive phenotype, historical significance, unusual mutation mechanisms, unique pathogenetic mechanisms, and specific inheritance patterns, like founder effects. This selection rationale is now explicitly mentioned in the updated table caption, offering a clearer understanding of the inclusion criteria for mutations listed.

Reviewer 1: l. 758-759; Combination of “address”, “therapies”, and “MG53…” seems illogical for “therapies” are used to improve the current status, and not equal to “MG53” or “steroids” (they are therapeutic reagents). Revising one or two expression(s) should be considered.

Author(s): This sentence has been updated to more accurately reflect the therapeutic approaches and their respective roles.

Reviewer 2 Report

Comments and Suggestions for Authors

The manuscript provides a good review of dysferlinopathies, mentioning various important aspects of these diseases. However, the review would benefit from including some graphic diagrams that represent the location of the protein in muscle tissue and its function therein. It would also be advisable to include a table that summarizes the therapies that exist for the treatment of these diseases.

Author Response

Response to Reviewer 2
Reviewer 2: The manuscript provides a good review of dysferlinopathies, mentioning various important
aspects of these diseases.
Author(s): Thank you for your positive feedback and insightful suggestions.
Reviewer 2: However, the review would benefit from including some graphic diagrams that represent the
location of the protein in muscle tissue and its function therein.
Author(s): In response to your suggestion, we have added a new figure illustrating the location of dysferlin
and its partner proteins in muscle cells.
Reviewer 2: It would also be advisable to include a table that summarizes the therapies that exist for the
treatment of these diseases.
Author(s): A new table summarizing various therapeutic approaches for Dysferlinopathies has been
incorporated into the manuscript.